# Evaluation of Teaching Signals for Motor Control in the Cerebellum during Real-World Robot Application

**DOI:** 10.3390/brainsci6040062

**Published:** 2016-12-20

**Authors:** Ruben Dario Pinzon Morales, Yutaka Hirata

**Affiliations:** 1Neural cybernetics laboratory, Department of Computer Science, Graduate School of Engineering, Chubu University, Kasugai 487-8501, Japan; rdpinzonm@ieee.org; 2Department Robotic Science and Technology, Graduate School of Engineering, Chubu University, Kasugai 487-8501, Japan

**Keywords:** cerebellum, motor learning, climbing fiber, robot control, neural network model

## Abstract

Motor learning in the cerebellum is believed to entail plastic changes at synapses between parallel fibers and Purkinje cells, induced by the teaching signal conveyed in the climbing fiber (CF) input. Despite the abundant research on the cerebellum, the nature of this signal is still a matter of debate. Two types of movement error information have been proposed to be plausible teaching signals: sensory error (SE) and motor command error (ME); however, their plausibility has not been tested in the real world. Here, we conducted a comparison of different types of CF teaching signals in real-world engineering applications by using a realistic neuronal network model of the cerebellum. We employed a direct current motor (simple task) and a two-wheeled balancing robot (difficult task). We demonstrate that SE, ME or a linear combination of the two is sufficient to yield comparable performance in a simple task. When the task is more difficult, although SE slightly outperformed ME, these types of error information are all able to adequately control the robot. We categorize granular cells according to their inputs and the error signal revealing that different granule cells are preferably engaged for SE, ME or their combination. Thus, unlike previous theoretical and simulation studies that support either SE or ME, it is demonstrated for the first time in a real-world engineering application that both SE and ME are adequate as the CF teaching signal in a realistic computational cerebellar model, even when the control task is as difficult as stabilizing a two-wheeled balancing robot.

## 1. Introduction

Cerebellar research in the last few decades has been accompanied by general consensus about its role in motor control, motor learning and motor coordination, yet abundant questions regarding the fundamental operations handled by this neural structure remain elusive [1]. One of the points of avid debate is the mechanisms of plasticity in the cerebellum [2]. In the original work by Marr-Albus [3,4], it was suggested that the cerebellum behaves as an adaptive filter in which mossy fiber inputs to the cerebellum were analyzed into sparse components at the granular cell layer and then conveyed by parallel fibers to the Purkinje cell. In this model, the location of plasticity corresponds to synapses between the numerous parallel fibers and the Purkinje cell, which change their strength according to a teaching signal conveyed by the climbing fiber (CF) input. Although a large repertoire of cerebellar models has been proposed since the early days of Marr, parallel fiber synapses remained the most commonly suggested location for cerebellar plasticity [5,6,7,8,9,10,11,12]. The debate pivots around the nature of the teaching signal in the CF.

Two types of movement error have been recognized as candidates for teaching signals: one from the sensory origin encoding error in movement kinematics (sensory error (SE)), the other from premotor systems encoding error in the motor command (motor error (ME)). For example, in the case of a tracking eye movement called smooth pursuit that is initiated when we try to follow a smoothly-moving visual target with the eyes, the movement error is the deviation of the eye velocity from the target velocity. This movement error causes a slippage of the target image on the retina that is detected by the retinal neuronal network. Thus, the output of the retina (ganglion cell activities) encoding the image slip kinematics is SE. Once the image slip is detected, the smooth pursuit system works to reduce it by modifying motor commands sent to the extra-ocular muscles. This modified motor command is ME. There are physiological evidence and computational studies supporting each teaching signal. In support of the notion that CF carry SE, CF activities highly correlated with retinal slip during optokinetic visual stimuli have been reported in rabbits [13] and monkeys [14]. In contrast, support for CF carrying ME has been provided by single-unit recording experiments using rabbits [15] and monkeys [16].

Theories have been built around these two types of teaching signals. On the one hand, the feedback-error-learning algorithm proposed by Kawato and colleagues [17,18] posits that plasticity in the cerebellum is driven by ME, which is made available at CF by reference structures outside the cerebellum that transform SE into ME. To produce an adequate teaching signal, these reference structures must implement a partial inverse model of the plant under control, which is sometimes troublesome when the control plant includes several degrees of freedom. Feedback-error-learning seeks to solve this problem by using a simple approximation to the inverse model, such as a gain term to transform SE into ME, which has been shown to work properly in simple control tasks and robotic applications [6]. For biological plants, it has been argued that with multiple-input multiple-output redundant systems, such as the vestibulo-ocular reflex in three dimensions, the connectivity needed by the reference structure becomes unfeasibly complex. On the other hand, Porrill and colleagues proposed the recurrent architecture for an adaptive filter in which the learning is driven by SE [5,19]. The structure of this algorithm partially follows the connectivity of particular cerebellar microzones, such as the flocculus; however, the role of granular and molecular layer interneurons has been unaccounted for. This recurrent algorithm allows stable adaptive learning using only observable SE signals and has been shown to work properly in complex robotic and simulation scenarios.

To solve the apparent dichotomy about the teaching signal in the CF, computational models of the cerebellum widely used in engineering applications may provide a potential framework to evaluate different CF contents and evaluated them in appropriate movement control tasks. However, there has not been any attempt so far. Here, we conduct a real-world testing of our cerebellar neuronal network model (CNN) in robot control and compare the performance of the robot when the different teaching signals are used. We employ a revised version of our CNN that incorporates a realistic cerebellar network architecture and a learning algorithm whose validity has been proven in both simulation and real-world experiments [9,11]. For this CNN, we configure adequate control tasks in which SE and ME are clearly dissociated. We demonstrate that SE and ME yield equivalent performance when the control task is simple, while they yield comparable performance even when the control task is more complex. Possible mechanisms underlying these results in the cerebellar neural network are discussed.

## 2. Materials and Methods

### 2.1. Neuronal Network Model of the Cerebellum

Inspired by the neuronal circuit of the cerebellar cortex, in particular that of flocculus, known to be involved in the horizontal vestibulo-ocular reflex (Figure 1A), we developed a neural network model of the cerebellum [9,11] (Figure 1B). As the cerebellar cortex has a homogeneous structure all over its volume, its basic learning algorithm is sometimes called a generic algorithm of the cerebellum [20]. We focused on the flocculus and the vestibulo-ocular reflex because they have been widely studied in anatomy and physiology [20]. The model comprises cell types found so far for which their physiological and anatomical properties are well understood [20]. Those are granule cells, Golgi cells, basket/stellate cells and Purkinje cells. We do not differentiate basket and stellate cells in our model. The model receives two types of input carried by mossy fibers and CF as in the real cerebellum. Mossy fiber inputs are postulated to carry desired motion signals, the efference copy of motor commands and sensory error signals (desired trajectory minus actual trajectory) [21,22,23]. These inputs innervate granule cells and Golgi cells. Golgi cells receive excitatory input from granule cells while projecting back to granule cells via inhibitory synapses, forming a negative feedback loop to regulate the activity of granule cells. Axons of granule cells, called parallel fibers, bifurcate and innervate basket cells and Purkinje cells via excitatory synapses. Basket cells project to Purkinje cells via inhibitory synapses, forming a negative feedforward pathway from granule cells to Purkinje cells. Purkinje cells are the sole output cell in the cerebellar cortex sending inhibitory projections down to the vestibular nucleus. This model is available for download at [24] or by accessing the model database of the cerebellar platform of the Japanese Neuroinformatics Node [25].

A proportional and derivative (PD) controller that is a feedback controller widely used in industry and other applications [26] is employed in tandem with the CNN model (Figure 1B), representing the non-cerebellar pathway in the vestibulo-ocular reflex neuronal circuit in Figure 1A as in previous simulation studies [27,28]. It provides the non-adaptive motor command to the control plant. The PD controller also generates the ME signal in this context since it is a simple gain term that approximates the inverse model of the control plant transforming SE into ME as proposed in Kawato’s feedback error learning scheme [6]. The parameters of the PD controller (proportional gain $k_{p}$ and derivative gain $k_{d}$) were designed following optimal settings for automatic controllers [29], so that the PD controller alone can stably operate the control plant during a simple desired motion (πsin(2π0.1t)).

The structure of our CNN model and its connectivity with the control plant, the non-cerebellar pathway and the vestibular nucleus remain the same disregarding the content of the error signal in our experiments. However, to be inline with the adaptive filter architecture proposed by Porrill et al. [19], our model would require the modification of the cerebellar structure so that the output of the Purkinje cell is projected to the input of the non-cerebellar pathway. We do not modify our cerebellar structure, maintain the architecture of the model as reported in the flocculus and only change the CF content.

### 2.2. Equations of the Model

The CNN model includes 755 granule cells, 5 Golgi cells, 15 basket cells and 1 Purkinje cell. These numbers preserve the ratio of the cell population and the convergence/divergence ratios of each cell type close to the cerebellar neuronal circuit [28] (shown in Table 1) and a sufficient number of granule cells for adequate control of the plants considered here. For a detailed study on the effects of scaling the number of neurons in the CNN and the output performance, refer to [30]. Each neuron model is described as follows:
(1)$$x_{\mathrm{GC}}=y_{\mathrm{MF}}\left(M_{\mathrm{MF}-\mathrm{GC}}\cdot W_{\mathrm{MF}-\mathrm{GC}}\right)+y_{\mathrm{GO}}\left(M_{\mathrm{GO}-\mathrm{GC}}\cdot W_{\mathrm{GO}-\mathrm{GC}}\right)$$
(2)$$y_{\mathrm{PF}}=\frac{1}{1+e^{-\sigma (x_{\mathrm{GC}}-\mu )}}$$
(3)$$x_{\mathrm{GO}}=y_{\mathrm{MF}}\left(M_{\mathrm{MF}-\mathrm{GO}}\cdot W_{\mathrm{MF}-\mathrm{GO}}\right)+y_{\mathrm{PF}}\left(M_{\mathrm{PF}-\mathrm{GO}}\cdot W_{\mathrm{PF}-\mathrm{GO}}\right)$$
(4)$$y_{\mathrm{GO}}=\frac{1}{1+e^{-\sigma (x_{\mathrm{GO}}-\mu )}}$$
(5)$$x_{\mathrm{BA}}=y_{\mathrm{PF}}\left(M_{\mathrm{PF}-\mathrm{BA}}\cdot W_{\mathrm{PF}-\mathrm{BA}}\right)$$
(6)$$y_{\mathrm{BA}}=\frac{1}{1+e^{-\sigma (x_{\mathrm{BA}}-\mu )}}$$
(7)$$x_{\mathrm{PC}}=y_{\mathrm{PF}}W_{\mathrm{PF}-\mathrm{PC}}+y_{\mathrm{BA}}W_{\mathrm{BA}-\mathrm{PC}}$$
(8)$$y_{\mathrm{PC}}=\frac{1}{1+e^{-\sigma (x_{\mathrm{PC}}-\mu )}}-0.5$$
where σ=8, μ=1/2, $x_{\mathrm{GC}}\in R_{+}^{1\times N_{\mathrm{GC}}}$ is the activity vector of all granule cells before being processed by the sigmoidal activation function $y_{\mathrm{PF}}$ (Equation (Equation 2)), $W_{\mathrm{MF}-\mathrm{GC}}\in R_{+}^{N_{\mathrm{MF}}\times N_{\mathrm{GC}}}$, $W_{\mathrm{GO}-\mathrm{GC}}\in R_{-}^{N_{\mathrm{GO}}\times N_{\mathrm{GC}}}$ are the matrix of synaptic weights between mossy fibers-granule cells and Golgi-granule cells, $M_{\mathrm{MF}-\mathrm{GC}}\in N^{N_{\mathrm{MF}}\times N_{\mathrm{GC}}}$, $M_{\mathrm{GO}-\mathrm{GC}}\in N^{N_{\mathrm{GO}}\times N_{\mathrm{GC}}}$ are masking matrices of zeros and ones that embed the convergence/divergence ratio for granule cells referring to Table 1. $x_{\mathrm{GO}}\in R_{+}^{1\times N_{\mathrm{GO}}}$ and $x_{\mathrm{BA}}\in R_{+}^{1\times N_{\mathrm{BA}}}$, $W_{\mathrm{MF}-\mathrm{GO}}\in R_{+}^{N_{\mathrm{MF}}\times N_{\mathrm{GO}}}$, $W_{\mathrm{PF}-\mathrm{GO}}\in R_{+}^{N_{\mathrm{GC}}\times N_{\mathrm{GO}}}$, $W_{\mathrm{PF}-\mathrm{BA}}\in R_{+}^{N_{\mathrm{GC}}\times N_{\mathrm{BA}}}$, $W_{\mathrm{PF}-\mathrm{PC}}\in R_{+}^{N_{\mathrm{GC}}\times N_{\mathrm{PC}}}$, $W_{\mathrm{BA}-\mathrm{PC}}\in R_{-}^{N_{\mathrm{BA}}\times N_{\mathrm{PC}}}$ follow the same notation for Golgi cells, basket cells and the Purkinje cell, and $N_{\mathrm{MF}},N_{\mathrm{GC}},N_{\mathrm{GO}},N_{\mathrm{BA}},N_{\mathrm{PC}}$ are the number of mossy fibers, Golgi, granule, basket and Purkinje cells, respectively; where $R_{+}=\left\{x\in R:0<x\leq 1\right\}$ and $R_{-}=\left\{x\in R:-1<x\leq 0\right\}$ for excitatory and inhibitory synapses, respectively. Firing rate vectors are all $R_{+}$, but $y_{C}$ is R={x:−0.5<x≤0.5}, so that it can be subtracted from the motor command produced by the PD and sent to the plant.

In order to standardize the contribution of each mossy fiber input to the cerebellar model, pre-cerebellar gains were set as normalizing scaling constants for each of them. These values were taken from the maximum values in the description of the robot hardware. Table 2 shows the scaling gains for each control plant. Then, the scaled mossy fibers were passed through an activation function ($y_{\mathrm{MF}}=1/1+e^{-\sigma (x_{\mathrm{MF}}-\mu )}$) to obtain firing rates compatible with the model (i.e., $R_{+}$), where *σ* and *μ* are as defined previously.

The learning rule regulated by the activity in the CF input that modifies the weights of parallel fiber-Purkinje cell synapses in the model is computed every sampling time according to the following equations:
(9)$$\begin{array}{cc} \Delta W_{\mathrm{PF}-\mathrm{PC}} & =-\gamma \cdot (y_{\mathrm{PF}}\left(t\right)-0.5)\cdot CF\left(t\right) \end{array}$$
(10)$$\begin{array}{cc} W_{\mathrm{PF}-\mathrm{PC}}\left(t+1\right) & =W_{\mathrm{PF}-\mathrm{PC}}\left(t\right)+\Delta W_{\mathrm{PF}-\mathrm{PC}} \end{array}$$
where $\Delta W_{\mathrm{PF}-\mathrm{PC}}$ is the matrix of changes in the synaptic weights of parallel fiber-Purkinje cell synapses, $W_{\mathrm{PF}-\mathrm{PC}}\left(t\right)\in R_{+}$ is the synaptic weight matrix in these synapses at sampling time *t* and *γ* is the learning rate. In this algorithm, when the signs of $\left(y_{\mathrm{PF}}\left(t\right)-0.5\right)$ and CF(t) are the same, the synaptic weights are decreased (long-term depression), whereas when their signs are different, the weights are increased (long-term potentiation). Note that the constant 0.5 operating $y_{\mathrm{PF}}$ is required to convert the $y_{\mathrm{PF}}$ firing rate from $R_{+}\to R$. Learning rate *γ* was set to be 0.008, which was chosen by considering a trade off between learning speed and convergence [9,11].

### 2.3. Control Plants

Two control plants, a brushed direct current motor and a two-wheeled balancing robot, are employed. The 2-W brushed motor (Figure 1C, RC-280SA, Mabuchi Motor, Chiba prefecture, Japan) generates a torque directly from the current supplied. The motor’s shaft is interfaced with an encoder circuit (ZMP Inc., Tokyo, Japan) for providing angular position information and a microcontroller board (e-nuvo CPU board, ZMP Inc., Tokyo, Japan) in charge of communication with the implementation computer via a serial interface. The mossy fiber inputs to the CNN model for this control are shown in Table 2. The PD controller for this plant is a position controller with $k_{p}=0.8$ and $k_{d}=0.01$ as the proportional and derivative constants, respectively. A virtual dynamical model simulation for this motor has been included in the repository of the CNN model as an example [24].

The two-wheeled balancing robot (Figure 1D, e-nuvo wheel, ZMP Inc., Tokyo, Japan) is an inverted pendulum system that is highly unstable and widely used in control engineering for testing control strategies [26]. It is equipped with a set of sensors, including a motor encoder and a gyroscope, which provide wheel angle (Figure 1D; ϕ(t)) and body tilt angle (Figure 1D; θ(t)), respectively. The robot is also equipped with a serial interface to allow communication with the computer (see its specification below) on which the model was implemented. The motion of the robot is driven by a single motor connected to the wheels of the robot that share the same shaft (single-input multiple-output system). The mossy fiber inputs for this control plant carry the signals described in Table 2. The PD controller in this control object is a parallel configuration of two controllers (body position controller: $k_{p}=5$ and $k_{d}=0.5$; and wheel position controller: $k_{p}^{*}=0.2$ and $k_{d}^{*}=0.05$) designed by following optimal settings for automatic controllers [26,29], so that the addition of both outputs (i.e., PD controller output) alone can stably operate the robot during a simple task ($\phi_{des.}\left(t\right)=\pi \sin\left(2\pi 0.1t\right)$, where $\phi_{des.}\left(t\right)$ is the desired wheel angular position).

The experiments were carried out using LabVIEW 2010 (National Instrument, Austin, TX, USA) running on a Windows computer (4 × 3.33 GHz Intel Core-i7 processor, memory: 16 GB) and communicating with the control plants via serial protocol at 57.6 kbps. The CPU board in both the motor and the two-wheeled balancing robot has a sampling time of $t_{s}=10$ ms that is the same time interval as used in the present configuration of the CNN model (for different setups and sampling times, see [31]).

### 2.4. Two Types of Error Information in CF

Sources of SE and ME, shown in Figure 1B, are generated from the non-cerebellar pathway. That is, the SE signal is computed from the difference between the desired and actual movement [20,32]. It carries position and velocity error components in kinematic coordinates (expressed in angle units: rad), whereas ME is computed as the output of the PD feedback controller, which represents error in motor command dynamic coordinates (expressed in electric current units: A) as proposed in the theory of the feedback-error-learning algorithm [6,17]. Additionally, the linear combination of the two types of signals (SE + ME) is also tested, as it has been suggested to be a plausible case in the real cerebellum [33]. The equations of SE and ME for each of the control plants are as follows:
(11)$$\begin{array}{cc} cc{\mathrm{SE}}_{\mathrm{DC}\;\mathrm{motor}} & =a_{1}\phi {\left(t\right)}_{e}+a_{2}{\dot{\phi }}_{e}\left(t\right) \end{array}$$
(12)$$\begin{array}{cc} {\mathrm{SE}}_{\mathrm{robot}} & =a_{1}\phi {\left(t\right)}_{e}+a_{2}{\dot{\phi }}_{e}\left(t\right)+b_{1}\theta_{e}\left(t\right)+b_{2}{\dot{\theta }}_{e}\left(t\right) \end{array}$$
(13)$$\begin{array}{cc} {\mathrm{ME}}_{\mathrm{DC}\;\mathrm{motor}} & =c_{1}PD\left(t\right) \end{array}$$
where subscript “e” denotes error and corresponds to the difference between the desired and yielded motion (e.g., $\phi_{e}\left(t\right)=\phi_{des.}\left(t\right)-\phi \left(t\right)$), constants $a_{1}\;=\;0.5$
${\mathrm{rad}}^{-1},a_{2}\;=\;0.02$ rad/${\text{s}}^{-1},b_{1}\;=\;5$
${\mathrm{rad}}^{-1},$
$b_{2}\;=\;0.5$ rad/${\text{s}}^{-1}$ and $c_{1}=-0.4$
${\text{A}}^{-1}$ are scaling values intended to equalize the contribution of each error component to the SE and ME signals so that all have the same importance. These values were calculated with the CNN model disabled (i.e., plant controlled only by the PD controller and desired motion $\phi_{des.}\left(t\right)=\pi \sin\left(2\pi 0.1t\right)$). ME in Equation (Equation 13) remained the same for both control plants. The combination of SE and ME is computed directly from Equations (Equation 11)–(Equation 13).

### 2.5. Performance Assessment Methodology

The desired motion $\phi_{des.}\left(t\right)$ for the control plants used to assess the performance of the cerebellar model is a sinusoidal motion. The sinusoidal wave was generated at frequencies ranging from 0.2 Hz–0.4 Hz and an amplitude of *π* (maximum angular velocity: 7.89 rad/s). The frequencies of the stimulus were chosen below the maximum controllable velocity of the robot (9.82 rad/s). The desired body tilt angle $\theta_{des.}\left(t\right)$ was set to zero degrees (90 degrees with the ground), so that the robot is commanded to remain vertical while following the desired wheel trajectory. Each stimulus was repeated up to 100 cycles. Considering that the random initialization of synaptic weights in the cerebellar neuronal network can be a source of variability in the performance of the model, five different initial sets of random synaptic weights were created for each set of weighted connections ($W_{\mathrm{PF}-\mathrm{PC}},W_{\mathrm{MF}-\mathrm{GC}},W_{\mathrm{MF}-\mathrm{GO}},W_{\mathrm{GO}-\mathrm{GC}},W_{\mathrm{PF}-\mathrm{GO}},W_{\mathrm{PF}-\mathrm{BA}}$, and $W_{\mathrm{BA}-\mathrm{PC}}$) sampled from a normal distribution in $R_{+}$ and $R_{-}$ with a standard deviation of 1 and a mean value of 0.5 and −0.5 for excitatory and inhibitory synapses, respectively. Special care is required to avoid over inhibition of the Purkinje cell when $W_{\mathrm{BA}-\mathrm{PC}}$ is large or instability in the Purkinje cell output when the feedback loop formed by $W_{\mathrm{PF}-\mathrm{GO}}$ and $W_{\mathrm{GO}-\mathrm{GC}}$ is large. Excluding $W_{\mathrm{PF}-\mathrm{PC}}$, all of the synaptic weights remained fixed during the experiments because they are considered the main synaptic plasticity locations of cerebellar motor learning [20,34]. The performance of the cerebellar neuronal network controller was measured as the root mean square error (RSE) of each control variable. In the case of the motor, there was one control variable (the shaft angle ϕ(t); Figure 1C), whereas there were two control variables for the two-wheeled balancing robot (wheel angle ϕ(t) and body tilt angle θ(t); Figure 1D).

## 3. Results

### 3.1. Simple Control Scenario

First, we tested our CNN model with different CF information contents in the easiest control scenario in our setup, that is the control of the angular position ϕ(t) of a metal shaft directly connected to a motor with a sinusoidal desired motion ($\phi_{des.}\left(t\right)=\pi \sin\left(2\pi 0.5t\right)$) (the experimental protocol is described in Materials and Methods). Three types of CF to the cerebellar model were tested separately; namely, CF carrying SE (Equation (Equation 11)), ME (Equation (Equation 13)) and the combination of the two, i.e., SE + ME. Figure 2 shows the behavioral consequences in control performance under each CF condition (SE, ME, SE + ME) in terms of the RSE of ϕ(t) (Figure 2A) and yielded shaft motions (Figure 2B). The performance obtained when the CNN model was disabled, i.e., the motor is controlled only by the PD, is included as “PD” in gray lines (Figure 2A,B). The RSE of ϕ(t) (Figure 2A) evidences the superior performance with the CNN model disregarding the CF content compared with the PD. The improvement was on average 0.015 rad or 30% of the initial value of 0.05 rad. Thus, using a CF with SE, ME or the combination of both produces similar performance in this simple control scenario. Not surprisingly, the temporal profiles of these CFs also look alike (Figure 7). Differences between the motion caused in the motor by the CNN model and that by the PD controller alone can be seen in the XY planes shown in Figure 2B. These XY planes are constructed by positioning the desired (X-axis) and yielded (Y-axis) motions in an XY plane rotated −45 degrees. In the ideal case, the desired and yielded motions are equal with a trajectory that lies in an horizontal line at y=0 (Figure 2B, dashed lines). Erroneous motions in the clockwise (CW) and counterclockwise (CCW) rotations of the motor shaft are mapped in the planes y>0 and y<0, respectively. To show the adaptation in the CNN model, the motions produced at Cycles #2, #50 and #95 are shown. Cycle #2 shows that the CNN model caused the motor to rotate in excess in the CW direction. This excess produced by the initialization conditions and the number of granule cells in the CNN model was corrected by Cycle #5. We have shown that increasing the number of granule cells brings robustness against initial conditions in a bi-hemispheric CNN [35]. XY planes corresponding to Cycles #50 and #95 show that the PD caused the motor to rotate in excess when transitioning from CW to CCW and from CCW to CW. In contrast, the CNN model caused the opposite effect in the shaft motion by reducing the rotation speed just before the transitions. As exemplified here, CF inputs carrying SE, ME or their combination are adequate error signals to drive plasticity at parallel fibers-Purkinje cell synapses in the CNN model and make little difference in the control performance if the control scenario is very simple (a shaft with 1 deg of freedom driven by a motor to follow a 0.5-Hz sinusoidal).

### 3.2. Difficult Control Scenario

After testing the easiest control scenario in our setup, the CF signals seemed to have similar temporal profiles, and thus, the relationship between the error signal used and the control performance attained is yet to be clarified. In this section, we tested our setup in a more difficult control scenario to elucidate this relationship.

The control scenario employs the two-wheeled balancing robot with sinusoidal desired motions for the wheel angle. The frequencies of the sinusoidal motion $\phi_{des.}\left(t\right)$ range from 0.2 Hz–0.4 Hz (the experimental protocol is described in Materials and Methods). Figure 3, in the same format as Figure 2, summarizes the control performance achieved when the frequency of the desired motion is 0.2 Hz. Similarly to the previous case, controlling the two-wheeled balancing robot with the CNN outperforms the PD controller alone. The wheel positions generated at Cycles #2, #50 and #95 are shown in Figure 3B. These trajectories evidence that the CNN progressively improved the yielded wheel motion so that the robot motion approached the desired motion ($\phi_{des.}\left(t\right)$). In contrast, the wheel motion generated by the PD shows considerably larger hysteresis, meaning that the yielded motion differs from the desired motion (Figure 3B, gray lines). Subtle differences can be seen in the control performance obtained by using SE, ME or their combination. Increasing the frequency of the desired motion from 0.2–0.3 and 0.4 Hz strengthened the differences between SE and ME. Figure 4 shows the temporal profile of the different CFs, the firing rate of the Purkinje cell and the average control performance at the three frequencies of $\phi_{des.}\left(t\right)$. The temporal profiles of CF carrying SE (Figure 4A, blue lines), ME (red lines) and SE + ME (green lines) show that at 0.2 Hz, all of the CFs look alike. However, increasing the frequency of $\phi_{des.}\left(t\right)$ reveals that ME is delayed with respect to the other types of CFs (Figure 4A, black arrows).

The firing rates of the Purkinje cell and the vestibular nucleus (Figure 4B) also evidence clear differences when using SE or ME in the CF. At 0.4 Hz, the firing rate of the Purkinje cell caused by ME and SE is out of phase and presents distinctive shapes. The outputs of the vestibular nucleus due to these contributions by the Purkinje cell also differ depending on the CF types (Figure 4B, black arrows). Figure 4C summarizes the control performance in terms of the average RSE of ϕ(t) at each one of the frequencies tested at the beginning (Cycle #5) and the end (Cycle #95) of the experiments. This figure shows that the performance with ME deteriorates (Figure 4C, red bars) as the frequency increases, to such an extent that at 0.3 and 0.4 Hz, there is no improvement over the initial error (Figure 4C, asterisks). On the contrary, the performance with SE always reduces the initial error and is better than using the PD or the combination SE + ME. Thus, SE is the best error signal in our setup during the control of a highly unstable two-wheeled balancing robot at high frequencies (>0.3 Hz).

### 3.3. Neural Consequences of the Different CF Types

In this section, we analyze the neural consequences of using SE, ME or SE + ME in the CNN model during the control scenario with the two-wheeled balancing robot. We show the adaptation produced at parallel fibers-Purkinje cell synapses, which are the only plastic locus in our model, and the correlation of mossy fibers, granule cells and the CF used. The correlation values are shown to investigate the information in the granule cells that, in combination with the CF input, resulted in the learned parallel fibers-Purkinje cell synaptic weights.

Figure 5 shows the 755 parallel fibers-Purkinje cell synaptic weights ($W_{\mathrm{PF}-\mathrm{PC}}$) during one of the experiments with CF carrying SE. $W_{\mathrm{PF}-\mathrm{PC}}$ with the other CF contents showed similar trends. The gray area in the figure shows the weights decreased by long-term depression from the initial value (black line). Those weights increased by long-term potentiation are shown above the black line. Some representative $W_{\mathrm{PF}-\mathrm{PC}}$ are presented in red lines. The number of potentiated and depressed weights varied depending on the types of CF. With CF carrying SE, ME and SE + ME, 65%, 71% and 68% of the $W_{\mathrm{PF}-\mathrm{PC}}$ weights were depressed, respectively. Across all of the experiments, the same trend was observed, namely the number of depressed synapses was larger by using ME than SE. The insets in Figure 5 show the normalized firing rate of the granule cells whose synaptic weights were the most potentiated (labeled as “a”) and most depressed (labeled as “b”) for each type of CF. The firing rate of potentiated granule cells are negatively correlated with the CF, whereas those depressed are positively correlated with the CF. Figure 6A shows the coefficient of correlation of the CF and the granule cell’s activity. The moving average (N = 13) is shown in bold lines. Granule cells have been sorted in the *x*-axis from the most depressed (*x* = 0) to the most potentiated (*x* = 755). The normalized synaptic weight of each sorted granule cell is shown (dashed line). Shadowed area shows those cells whose activity was depressed. The coefficient of correlation when using SE, ME or SE + ME shows that those granule cells potentiated are negatively correlated with the CF, meaning that increasing their activity reduces the error signal, whereas those GCs depressed are positively correlated with the error signal.

The correlation of the granule activities with the mossy fiber inputs reveals intrinsic differences when using SE (blue lines) and ME (red lines). Figure 6B shows the coefficient of correlation of the mossy fiber carrying wheel position error ($\phi_{e}\left(t\right)$) and the activities of granule cells. In this correlation, those granule cells positively correlated were preferably potentiated when using SE, contrary to when using ME. Figure 6C shows the coefficient of correlation of the mossy fibers carrying wheel desired position ($\phi_{des.}\left(t\right)$) and the activities of granule cells. In this correlation, those granule cells negatively correlated were preferably potentiated when using SE, contrary to when using ME. These two correlations evidence intrinsic differences when using SE and ME in our setup. Granule cells with firing activity in-phase with error components (i.e., $\phi_{e}\left(t\right)$ and ${\dot{\phi }}_{e}\left(t\right)$) and those out of phase with desired motions (i.e., $\phi_{des.}\left(t\right)$ and ${\dot{\phi }}_{des.}\left(t\right)$) were preferably engaged to produce the Purkinje cell activity when using SE. When using ME, the opposite relationship was more prominent among the granule cells potentiated. Other mossy fibers, such as the efference copy, did not show differences between SE and ME. Employing the combination of SE and ME seems to soften the correlations observed with SE and ME to an intermediate point (Figure 6, green lines); therefore, it is not a surprise that using SE + ME as CF produces intermediate control performance (Figure 4, green lines).

## 4. Discussion

Debate about the error information encoded in the CF input to the cerebellum has been going on for several years [2,5,16,36]. Two types of error have been greatly defended: SE and ME. Behavioral and neurophysiological support has been presented for SE in rabbits [13] and monkeys [14] and similarly for ME in rabbits [15] and monkeys [16]. Consensus has been difficult to reach because these behavioral and neurophysiological experiments employ simple movement tasks where the temporal patterns of SE and ME are alike. To the best knowledge of the authors, this study is the first direct comparison of the types of error information encoded in the CF input to a CNN and with the motor performance attained during a real-world engineering application. Our setup allowed us to configure the difficulty of the control task, change the type of error encoded by the CF and evaluated the yielded control performance, thus enabling us to effectively disassociate SE and ME.

### 4.1. Type of Error Signal and the Yielded Robot Control Performance

Our experimental results showed that both SE and ME, despite producing unique behavioral and neural changes in the CNN model, especially at the Purkinje cell output (Figure 6B), are adequate error signals to govern successfully the plants in a diversity of control tasks (simple and complex task) (Figure 2, Figure 3, Figure 4, Figure 5 and Figure 6). The practical implications of these results are that under certain experimental setups, investigating the consequences in motor learning when using motor or error information in the CF might yield positive, but inconclusive results. Thus, the proper selection of control plants and stimuli that allow the dissociation of motor and control error contents is imperative. This is the first direct demonstration suggesting that both SE and ME can be an adequate error signal to teach a realistic cerebellar computational model in real-world engineering applications. These results are in agreement with biological evidence from the horizontal vestibulo-ocular reflex (VOR) and optokinetic reflex (OKR) systems, a feed-forward and a feedback system sharing the same controller (i.e., cerebellar microcomplex) and the same control object (i.e., eye plant) with CF carrying SE and ME, respectively [2]. However, further evaluation using the two-wheeled balancing robot with sinusoidal desired motion at different frequencies revealed quantitative differences in control performance caused by SE and ME (Figure 6). Namely, SE yielded better control performance than ME (Figure 6C). This might be trivial because we evaluated the control performance in terms of the RSE of SE (i.e., RSE of ϕ(t)), and the CNN model was trained to reduce SE. Since the ultimate goal of a control task is to reduce the error between the desired motion and the yielded motion (i.e., reduce SE) in biological systems and engineering applications, we employed the RSE of SE as a measure of the goodness of the control performance. In addition, SE signals are observable. In the case of ME, the desired motor commands are not observable, and therefore, the performance cannot be computed directly in terms of ME.

The correlation of the firing rate activity of granule cells, CF and mossy fibers also revealed intrinsic differences caused by using SE, ME or their combination (Figure 6). In particular, granule cell activity preferably potentiated by LTP at parallel fibers-Purkinje cell synapses with SE and ME differed in the correlation with the mossy fiber inputs. This means that to produce the Purkinje cell activities shown in Figure 4 with SE and ME, granule cells with different mossy fiber inputs were engaged, thus the differences in control performance. In particular, granule cells carrying error information were preferentially potentiated by sensory error, and granule cells carrying desired motions were preferentially potentiated by motor error. In our setup, the relation between granule cells activity and the CF content holds true for the two control plants used; however, extrapolation to other systems is not evident since changing the number of mossy fiber inputs and the size of the population of granule cells would result in a different mapping of inputs to parallel fibers, which in turn also affects the information that reaches the Purkinje cell. Clarifying this relation requires further study of the mossy fiber inputs to each granule cell and their correlation with the CF error content. For a study in this direction with a bi-hemispherical version of the CNN, refer to [31].

We have employed a PD controller as the non-cerebellar pathway that produces ME not only because it is one of the most flexible, effective and popular feedback controllers [26,29], but because it has been proposed as a simple approximation to the inverse model of the controlled plant in the feedback-error-learning theory [18,37,38]. The PD may be actually physiologically plausible [19] since its computation requires sensory information and its first derivative (e.g., eye velocity and eye acceleration), which are available at brain stem [39] and the cerebellum [40]. However, as the complexity of the plant increases, this approximation is deficient, as we have shown in the experiments with the two-wheel balancing robot (Figure 6). We observed that increasing the frequency of the desired motion (i.e., increasing the complexity of the control task) produced a delayed ME signal, and performance weakened (Figure 6A). This is produced by the frequency characteristics of the PD controller and the fact that it is not a close approximation to the inverse model of the two-wheeled balancing robot. These results confirm a major drawback of using ME as the teaching signal in a realistic cerebellar model as proposed in the feedback-error-learning theory since the complexity of the reference structures transforming SE into ME must increase with the complexity of the control plant, which could become troublesome in highly elaborated control scenarios.

Our results also showed that a linear combination of SE and ME encoded in the CF is an adequate error signal in our CNN model to control the plants successfully. The performance attained was always better than when the CNN model was disabled (Figure 3, Figure 4 and Figure 5, magenta lines). These results are in-line with physiological evidence in monkeys during arm reaching experiments that has suggested that CF encodes both SE and ME at different times of the arm movement [33], and thus, CF carries the error signal required for real-time control and learning of movements.

### 4.2. Generalization and Limitation of the Current Results

The control performance with the CNN model was stable and consistent while considering a basic control plant (DC motor) and one of the most challenging plants in control engineering (the two-wheeled balancing robot) [26], combined with simple and complex stimulus (low and high frequency sinusoidal desired motion) within the operation boundaries of our setup (real-time operation at a sampling frequency of 10 ms). The characteristics of the control scenarios and our setup made it difficult to reproduce the delay of 100 ms (10× the sampling frequency) shown in the visual inputs in the real cerebellum [34], which correspond to ϕ(t) and θ(t) in our model. However, under different circumstances, we believe our CNN model would be able to learn and produce adequate motor commands. These control plants have one and two degrees of freedom, respectively. Therefore, we may not be able to simply extend the current results to those with higher numbers of degrees of freedom. There is a series of evidence in cerebellar control of eye movement showing that different zones in cerebellar flocculus control different single axes of rotation of the eye (i.e., yaw and pitch rotations) [41]. Thus, our current setup to control objects is inline with this view. To apply the model to controlling other objects with higher numbers of degrees of freedom, we may employ the same number of cerebellar models for controlling each axis of motion [35]. Evaluating the nature of error signals suitable for this situation is in our scope of future studies.

In the current configuration of our CNN model, we have only considered plasticity at parallel fibers-Purkinje cell synapses, which are regarded as the loci of fast adaptation in the cerebellum [42], as a minimum requirement for the motor learning algorithm [2,20]. Multiple sites of plasticity in the cerebellum have been reported, and their involvement in motor learning has been argued [10,43,44]. Therefore, evaluating the effects in motor performance during robot control when the CNN model includes multiple sites of plasticity remains a future study of our CNN model.

### 4.3. Comparison with Other Cerebellar Models

Computational models of the cerebellum and their successful interpolation into engineering applications have extensively been reported. In general, these models include a form of synaptic plasticity at parallel fibers-Purkinje cell synapses driven by the CF input as presented in our CNN model, and that corresponds to the basis for cerebellar motor learning. Examples of CF encoding SE include models used for simulation of the eye plant [45], control of pneumatic muscles [46] and control of robotic arms [10,47]. Similarly, examples of cerebellar models with ME in their CF input include robotic arms [18] and inverted pendulum systems [11,38]. We have demonstrated that the CF input in our CNN model can be configured to include SE or ME information for driving plasticity at parallel fibers-Purkinje cell synapses and control the robot plant successfully. To the best knowledge of the authors, this study is the first direct comparison of the type of teaching signal encoded in the CF input to a cerebellar model and with the motor performance attained during a real-world engineering application. In contrast to spiking models of the cerebellum [7,12,48,49], due to the level of abstraction in our CNN model (i.e., firing rate neuron models), spike patterns and temporal or spatial effects were impossible to evaluate. This would require the construction of a cerebellar network with spiking neuronal models that could endanger the real-time real-world application in control engineering and a network with realistic physical properties [28]. Nonetheless, large-scale cerebellum-like spiking model has also been demonstrated to run in real time [12], and its application into robotics has been suggested. We have also extended our CNN model to a bi-hemispherical configuration (biCNN) that presents additional benefits and reproduces a form of cerebellar asymmetrical motor adaptation using SE information in the CF input [30].

## 5. Conclusions

The current study demonstrated for the first time in a real-world engineering application that both sensory error (SE) and motor error (ME) are adequate as the climbing fiber teaching signal in a realistic computational cerebellar model not only when the control task is simple, but also when the control task is as difficult as stabilizing a two-wheeled balancing robot.

## Figures and Tables

**Figure 1 brainsci-06-00062-f001:**
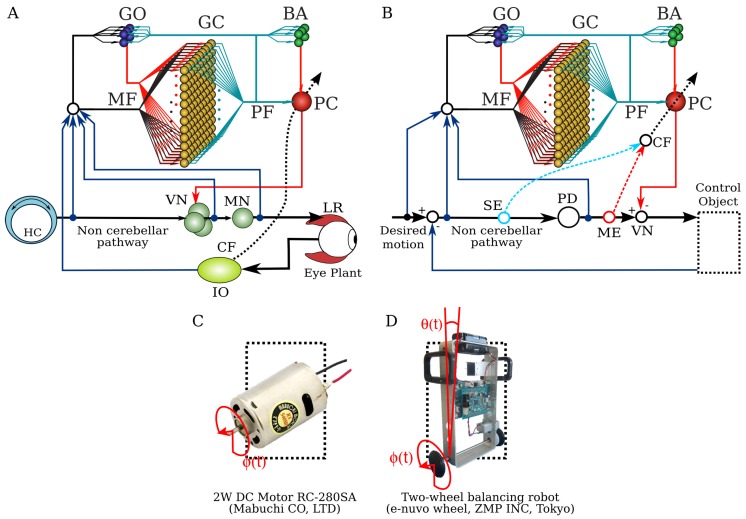
Neural circuit of the horizontal vestibulo-ocular reflex (VOR) and cerebellar neuronal network (CNN) model. (**A**) The neural circuit of the horizontal VOR. CF, climbing fiber; GO, Golgi cell; GC, granule cell; HC, horizontal canal; IO, inferior olive; LR, lateral rectus muscle of the eye; MF, mossy fiber; MN, motor neuron; BA, basket/stellate cell; PC, parallel fibers; VN, vestibular excitatory/inhibitory neurons. Blue and red lines indicate excitatory and inhibitory action, respectively. The dashed line shows the CF. (**B**) The structure of the CNN model with a proportional and derivative (PD) feedback controller that represents the non-cerebellar pathway depicted in (A). Labeled dashed lines show the origin of the sensory error (SE) (blue dashed line) and motor error (ME) (red dashed line) sources of CF, respectively. (**C**) A 2-W direct current motor with 1 degree of freedom (DOF). The control variable is shaft angular position ϕ(t). (**D**) Two-wheeled balancing robot with 2 DOFs. The control variables are wheel angular position ϕ(t) and body tilt angle θ(t).

**Figure 2 brainsci-06-00062-f002:**
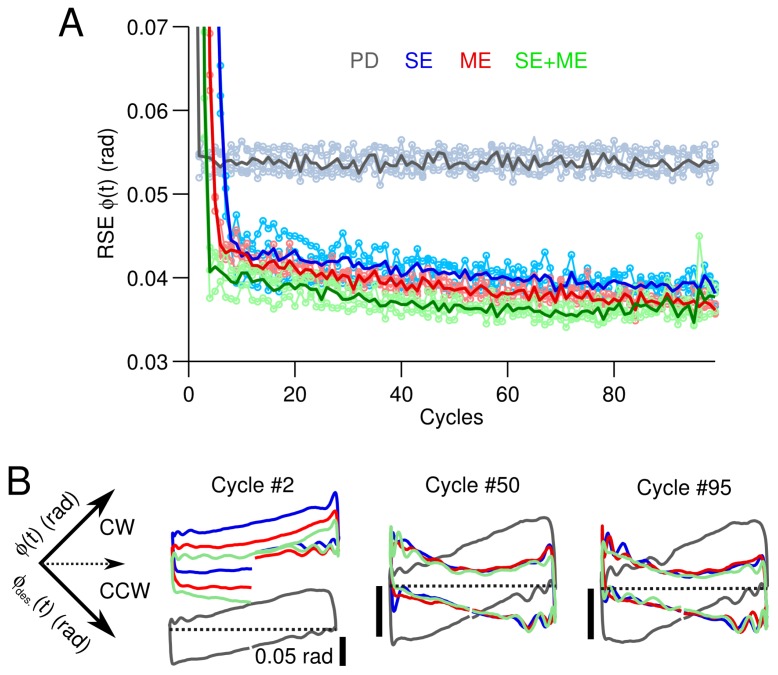
Performance during the control of the direct current motor with sinusoidal desired motion. (**A**) Control performance in terms of the root mean squared error (RSE) of ϕ(t) for the different types of climbing fiber (CF) averaged across the five experiments with different sets of initial synaptic weights. The mean value (thick line) and raw performance are shown. For reference, the performance with the proportional and derivative (PD) controller alone is shown (gray lines). (**B**) Yielded (ϕ(t), Y-axis) versus desired ($\phi_{des.}\left(t\right)$, X-axis) shaft angle position (rotated −45 degrees) when the different CFs are employed. The trajectories at Cycles #2, #50 and #95 are shown. Black scale lines are 0.05 rad.

**Figure 3 brainsci-06-00062-f003:**
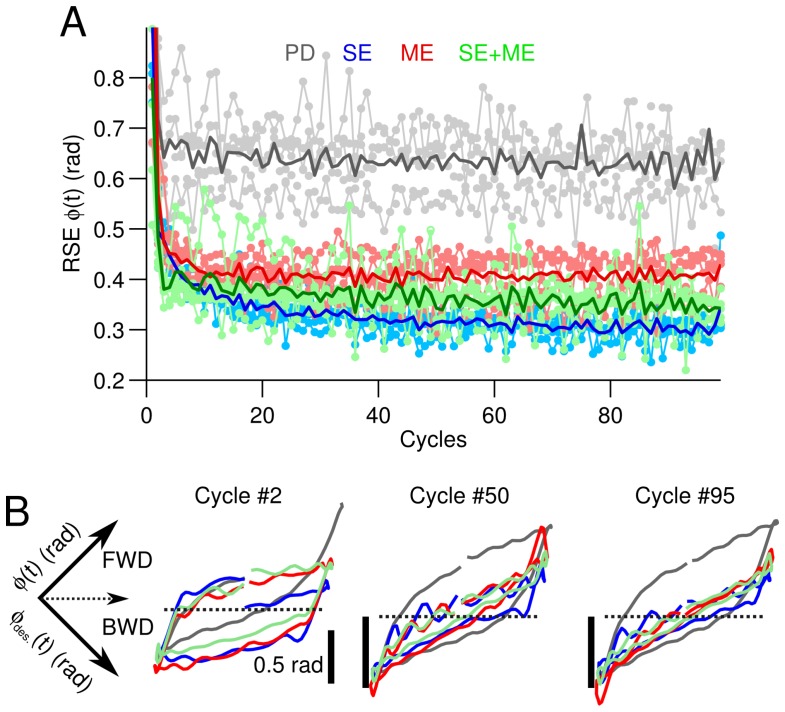
Performance during the control of the two-wheeled balancing robot with sinusoidal desired motion. (**A**) Control performance in terms of the root mean squared error (RSE) of ϕ(t) for the different types of climbing fiber (CF) averaged across the five experiments with different sets of initial synaptic weights. The mean value (thick line) and raw performance are shown. For reference, the performance with the proportional and derivative (PD) controller alone is shown (gray lines). (**B**) Yielded (ϕ(t), Y-axis) versus desired ($\phi_{des.}\left(t\right)$, X-axis) wheel angle position (rotated −45 degrees) when the different CFs are employed. The trajectories at Cycles #2, #50 and #95 are shown. Notation as in Figure 2. Scaled black lines mark 0.05 rad.

**Figure 4 brainsci-06-00062-f004:**
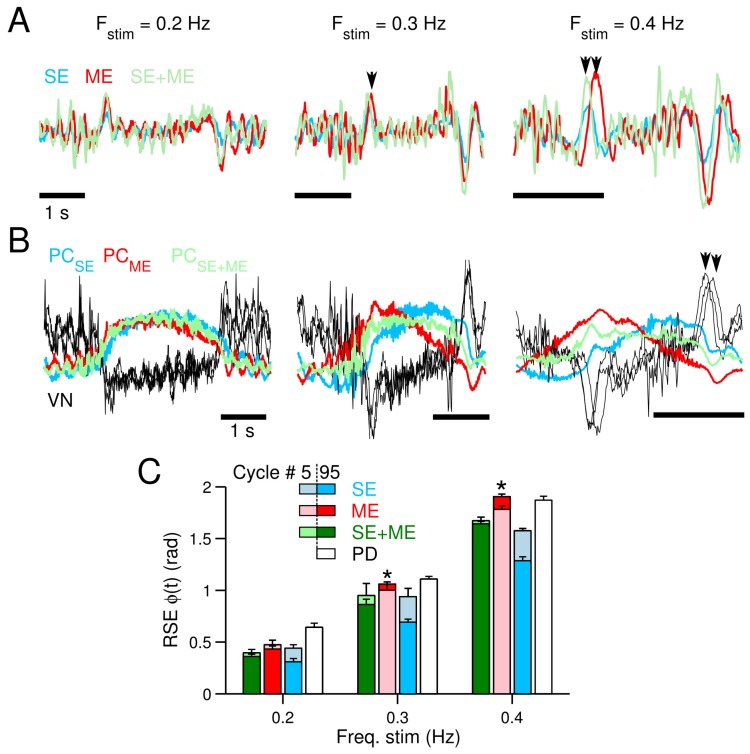
Firing rates of climbing fiber (CF) and the Purkinje cell (PC) during the control of the two-wheeled balancing robot. Three frequencies of $\phi_{des}\left(t\right)$ are shown, 0.2, 0.3 and 0.4 Hz. (**A**) CF activity when carrying sensory error (SE) (blue), motor error (ME) (red) and SE + ME (green). Arrows show that CF carrying ME is delayed with respect to the other types of CF. (**B**) Firing rate of the Purkinje cell and vestibular nucleus (VN, black) at the different frequencies tested. Arrows show that the vestibular nucleus activity produced with CF carrying SE and ME is clearly different at 0.4 Hz. (**C**) Averaged control performance in terms of the root mean squared error (RSE) of ϕ(t) at Cycles #5 (light colors) and #95 (dark colors) at the different frequencies of the desired motion.

**Figure 5 brainsci-06-00062-f005:**
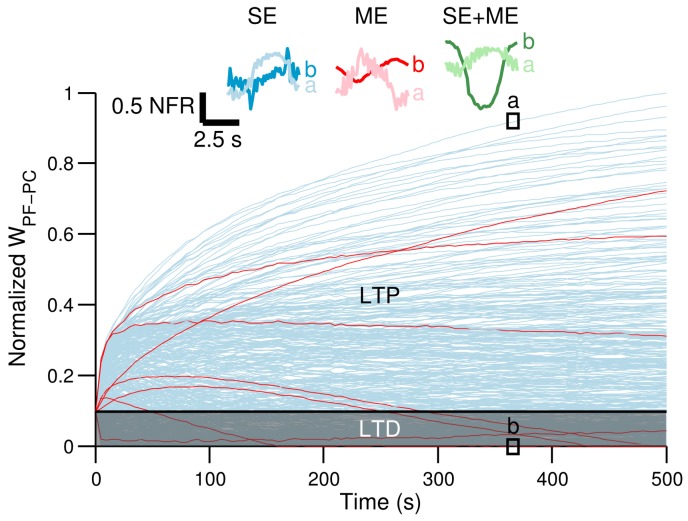
Synaptic weights between granule cells and the Purkinje cell ($W_{\mathrm{PF}-\mathrm{PC}}$) during the control of the two-wheeled balancing robot. Red lines show some examples of $W_{\mathrm{PF}-\mathrm{PC}}$. The horizontal line separating the shadowed areas of long-term depression (LTD) and long-term potentiation (LTP) is the initialization synaptic weight. Insets show the normalized firing rate (NFR) of granule cells whose $W_{\mathrm{PF}-\mathrm{PC}}$ synapses were predominately potentiated “a” or depressed “b” when climbing fiber (CF) carried sensory error (SE) (blue lines), motor error (ME) (red lines) or the combination of both (green lines).

**Figure 6 brainsci-06-00062-f006:**
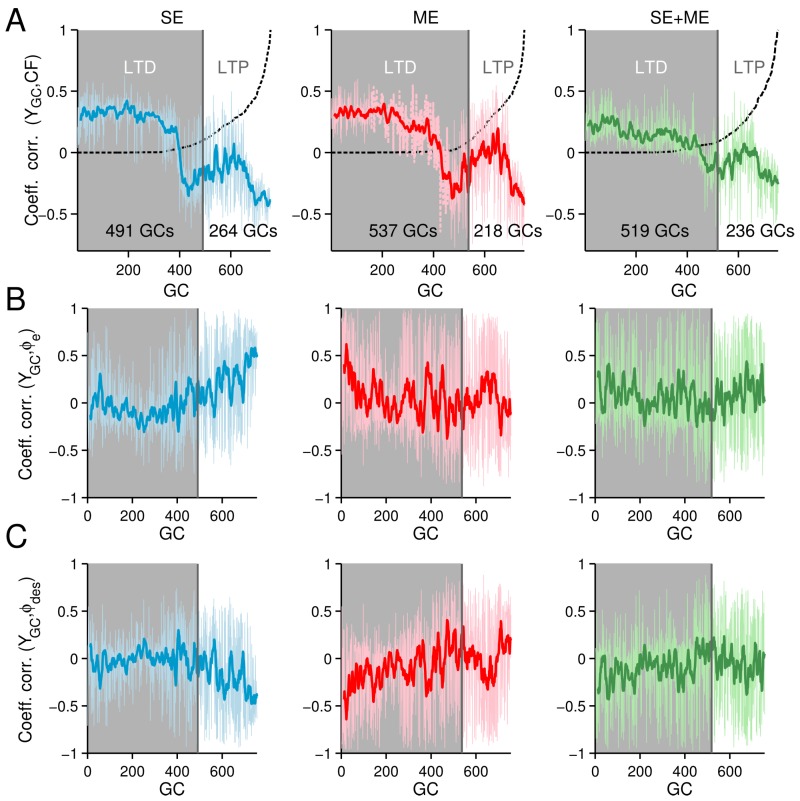
Correlation of granule cell (GCs) firing rate, climbing fiber (CF) and mossy fibers (MFs) in the CNN model. The granule cells in the x-axis have been sorted according to their $W_{\mathrm{PF}-\mathrm{PC}}$ value, so that at the origin is the most depressed and at *x* = 755 is the cell most potentiated during the control of the two-wheeled balancing robot. Shadowed areas show the granule cells depressed. The moving average (*N* = 13) is shown in bold colors. The synaptic weight is shown with a dashed lined. (**A**) Coefficient of correlation between the CF and granule cells; (**B**) coefficient of correlation between the mossy fiber carrying wheel position error ($\phi_{e}\left(t\right)$) and the granule cells; (**C**) coefficient of correlation between the mossy fiber carrying desired wheel position ($\phi_{des}\left(t\right)$) and the granule cell.

**Table 1 brainsci-06-00062-t001:** Convergence and divergence synaptic ratio in the CNN (cerebellar neuronal network) model.

	No. of Cells	Divergence	Convergence
Mossy fibers (MF)	7/5 *		
Golgi (GO)	5		
Granule (GC)	755		
Basket/stellate (BA)	15		
Purkinje (PC)	1		
Parallel fibers (PF)	755	
MF → GC		1:150	4:1
MF → GO		1:5	50:1
GC (parallel fibers)→ GO		1:4	150:1
GO → GC		1:600	3:1
GC (PF)→ BA		1:30	50:1
GC (PF)→ PC		1:1	755:1
BA → PC		1:1	15:1

* Number of mossy fiber inputs to the cerebellar model changes with the control plant; seven for the two-wheeled balancing robot, five for the motor in the present study.

**Table 2 brainsci-06-00062-t002:** Characteristics of the control plants.

Object	Outputs (Sensors)	Mossy Fibers	Scaling Gain
Motor	ϕ(t): shaft ang. pos. (rad)	1-des. shaft ang. pos.	0.1 ${\mathrm{rad}}^{-1}$
	$\dot{\phi }\left(t\right):$ shaft ang. vel. (rad/s)	2- des. shaft ang. vel.	0.19 s/rad
		3- shaft ang. pos. error	0.5 ${\mathrm{rad}}^{-1}$
		4- shaft ang. vel. error	0.07 s/rad
		5- efference copy	1 ${\text{A}}^{-1}$
Robot	ϕ(t) wheel angle (rad)	1- des. wheel ang. pos.	0.03 ${\mathrm{rad}}^{-1}$
	$\dot{\phi }\left(t\right)$ wheel ang. vel. (rad/s)	2- des. wheel ang. vel.	0.04 s/rad
	θ(t) body tilt ang. pos. (rad)	3- body tilt ang. pos. error	1 ${\mathrm{rad}}^{-1}$
	$\dot{\theta }\left(t\right)$ body tilt ang. vel. (rad/s)	4- body tilt ang. vel. error	0.5 s/rad
		5- wheel ang. pos. error	0.1 ${\mathrm{rad}}^{-1}$
		6- wheel ang. vel. error	0.2 s/rad
		7- efference copy	0.5 ${\text{A}}^{-1}$

Angular (ang.); desired (des.); velocity (vel.); position (pos.).

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
