# Peer review of "Evaluation of Teaching Signals for Motor Control in the Cerebellum during Real-World Robot Application"

_brainsci, 2016, doi:10.3390/brainsci6040062_

Round 1

Reviewer 1 Report

Manuscript: Evaluation of teaching signals for motor control in the cerebellum during real-world robot application 

General Comments

The manuscript addresses the still open question of whether the cerebellar climbing fiber system encodes sensor error (SE) or rather motor error (ME) as the teaching signal for the Purkinje cells. In contrast to previous theoretical and simulation studies, a real-world engineering approach is used with experimental positioning tasks performed by an adaptive cerebellar-like controller driving two different mechanical plants: a simple motor-activated shaft and a more complex two-wheel robot. The bio-inspired controller always performed better than the engineering gold standard: the proportional-derivative (PD) controller. With the simple motor-driven shaft, there was no difference between SE, ME or SE+ME. With the two wheel robot as the plant, encoding of SE in the climbing fibers was slightly better than ME encoding, but both, and also the combination of the two (SE+ME), could adequately conrol the robot.

With a correlation analysis of the neural signals in mossy fibers, granule cells, and parallel fibers the manuscript convincingly shows that both SE and ME are adequate teaching signals even in difficult control tasks. Even though this is no direct neurophysiological evidence for the encoding of both SE and ME by cerebellar climbing fibers, the manuscript contributes to our knowledge by providing support for this hypothesis.

Due to the ordering of the chapters and the use of many abbreviations, however, the manuscript is very difficult to read.

Minor Comments

The readability of the manuscript is hampered by the following:

1) The sequence in which the Methods are presented. It is confusing to assume many part of the Results and Discussion that the methods have already been presented. I therefore suggest to reorder the manuscript such that essential parts of the model are presented in the Introduction. This is a valid approach, because the model has already been introduced in a previous paper.

I particular, I would transfer section 4.1 from lines 277-302 together with Figure 6, but without Table 1, to an additional section after line 45 in the Introduction. In addition to the exemplary description of the difference between SE and ME in lines 35-41, I would also refer the reader to the model (now in Figure 1) and to the labels SE and ME before and after the PD, respectively.

2) Abbreviations

For the reader, the abundant use of abbreviations is very hard to decode and, therefore, confusing. In my view, only the use of CF, SE, ME, PD, and CNN are meaningful. The long list of all other abbreviations (PC, GC, GO, PF, DC, CW, CCW, VN, LTD, LTP, VOR, OKR, MF, BA, UART, SIMO, CPU) should be avoided in the text. They can, however, be used in the figures and tables, if defined in one of the legends.

For example, "DC motor" can become "motor" and "UART chip" (not "USART"!) can become "serial interface". PF should always be addressed as "parallel fibers" etc.

3) The colours of the neural connections in the model of Figure 6 are confusing. For example, it is confusing to see the inhibitory output of Purkinje cells in red instead of blue.

4) Figure ordering:

Move Figure 1 from line 127 to line 79 (does the scaling of 0.05 rad in B change from #2 to #50 to #95? This is confusing)
Move Figure 2 from line 137 to line 105 (does the scaling of 0.5 rad in B change from #2 to #50 to #95? This is confusing)
Move Figure 4 from line 162 to line 143
Move Figure 5 from line 176 to line 149

5) Late definitions of abbreviations: Some abbreviations are used before being defined. By reordering of the manuscript (see Minor Comment 1) and by avoiding abbreviations (see Minor Comment 2), the confusion which this creates might be reduced.

5) The Baskett Cell loop is shown in the model, but its significance for the controller function is not discussed. The authors might want to add some explanation. The stellate cells are only mentioned (in line 285), but never used. If not used, this should be explicitly mentioned.

Specific Comments

P1, L29-30: Replace "are the most common location" by "remained the most commonly suggested location"

P2, L33: Replace "sensory output" by "sensory input" (a reader might consider the cerebellum as the central part of the model, which would require the sensory information to be defined as an input to and not an output of the cerebellum).

L38: "output of the retina" by "input from the retina" (the same considerations as above apply)

L 42: "carries" by "carry"

L 46: "has been build" by "have been built"

L46: "In one hand" by "On the one hand", L56: "In the other hand" by "On the other hand"

L49: "transforms" by "transform" and "references" by "reference"

L50: "witch" by "which"

L51: "included" by "includes"

L53: "works" by "work"

L65: "thus far" by "so far"

L68: "realistic" by "a realistic" and "learning algorithm" by "a learning algorithm"

P3, L81-81: Replace "(SE, blue lines; ME, red lines; SE + ME, green lines)" by "(SE, ME, SE + ME)". The colour coding should be described in the legend of Figure 1 instead of the text.

L86: "was in average" by "was on average"

L97: "GC" is used here without a previous definition in the text; it is, however, defined in the Abstract.

L 118: "On the contrary" by "In contrast"

P 4, Legend of Figure 1:
a) RSE is used without being defined before; it is defined much later in line 388 (Please see above comment with regard to reordering the manuscript).
b) "the five experiments": At this point it is not clear which experiments are meant.

L 128: "VN" by "vestibular nucleus" (VN is defined much later in line 293). However, in the legend of Figure 6) it is defined as "vestibular neuron". Please correct the legend accordingly since a (peripheral) vestibular neuron doesn't perform an operation as in Figure 6 B.

L134: "to such extend" by "to such an extent"

L135: "On the contrary" by "In contrast"

P5, Legend of Figure 2: "the five experiments": see above comment on Legend of Figure 1

L148: "LTD" by "long-term depreesion" and "LTP" by "long-term potentiation" (LTP, LTD are defined much later in lines 330-331)

P6, Legend of Figure 3: VN undefined

P 9, L 191: Replace "specially" by "especially"

L193: "Thus confirming" by "This confirms"

L 197: "VOR" and "OKR" used without a previous definition; see above Minor Comment 2

L 203: Replace "RSE" by "root mean squared error (RSE)"

L213-215: As this sentence is hard to understand, I suggest using the following ("respectively" is very confusing):
"In particular, granule cells carrying error information were preferentially potentiated by sensory error and granule cells carrying desired motions were preferentially potentiated by motor error."

L216: "PD" used before its definition. This might change after implementing Minor Comment 1.

L227: "results confirms" by "results confirm"

L227: "as teaching" by "as the teaching"

L244: "in our model" is yet unclear because the model has not been introduced, yet. This might change with Minor Comment 1.

L309: "detail study" by "detailed study"

L310: "reefer" by "refer"

L339: Replace "USART" by "UART"
or even better:
"via USART Serial protocol" by "via a serial interface"

L348: "equipped with a USART chip" by "equipped with a serial interface"

L351: Delete "SIMO"

L384: "PC cell" by "Purkinje cell"

Figure 6:
Show signals Phi(t) and Theta(t)

Table 2:
In the legend, refer the reader to the signals Phi(t) and Theta(t) from Figure 6

Author Response

First and foremost, we would like to thank the reviewers for their valuable and detailed comments on our previous manuscript. We addressed each and every major and minor issue that they raised to improve our paper as listed below:

1) The sequence in which the Methods are presented.

We have changed the order of the sections. In the updated version of the manuscript the section describing the model, Methods and Materials, is presented right after the introduction and before the results. In this way all the abbreviations, measures of performance, and features of the experimental setup are introduced before the results. We believe that this improves greatly the flow of the paper as the reviewer suggested in his/her comment.

2) Abbreviations

In connection to the point above, we have reduced substantially the abbreviations and only SE, ME, CNN, CF are being used in the updated manuscript.

3) The colours of the neural connections in the model of Figure 6 are confusing.

Thank you for pointing out this color mismatch. We updated the caption in Figure 6 to “Blue and Red lines indicate excitatory and inhibitory action, respectively”, which is consistent with the colors used in the Figure.

4) Figure ordering:

Due to the change in the sections, as mentioned in 1), the figures are presented in an order that facilitates the comprehension of the materials used before the discussion of the results.

5) Late definitions of abbreviations:

This has been solved in 1) and 2) above so that no abbreviation is presented later than the section Materials and Methods.

5) The Basket Cell loop is shown in the model, but its significance for the controller function is not discussed.

This is a very critical and interesting point that we would like to explore by taking more time and effort in a future study as an extended study of the current research. Indeed this loop, which has also been reported to include bidirectional plasticity, could modulate directly the activity of the Purkinje cell and motor learning/coordination. Regarding Stellate cells in the molecular layer, we have decided to combine Basket and Stellate cells into a single abstraction (Basket cells) because their anatomical input and output connections are basically the same, and thus their roles in cerebellar signal processing could be assumed to be the same. Nonetheless, differentiating these cells and studying their effects on the Purkinje cells activity would be an interesting pursue for a future computational study once enough amounts of evidence are available for modeling these neuron types separately and precisely.

6) Specific comments

We have amended all of the specific comments and language suggestions. In our updated manuscript we have marked the changes in red to facilitate its reading. Thank you again for your kind and very careful reading.

Reviewer 2 Report

This is a well-written and clearly presented paper. The authors build on previous work with a detailed computational model of the cerebellum. Here they aim to bring in empirical evidence to the debate about the nature of error signals on cerebellar motor learning. The introduction, presentation of the results and discussion are appropriate. Hence, methodologically the paper is sound. In my opinion this work deserves publication, even though the significance of the results may be limited because of the concerns that I will expose below.

MAJOR COMMENTS

The authors provide essentially negative evidence: they “fail” to demonstrate a major effect of using one type of error signal or the other in their setups. Hence, all they can affirm is that in the setups they have devised, this difference doesn’t exist. But, what would have been the outcome in a more complex task? Note that the definition of complexity in the task is rather arbitrary. In both the “simple” and the “complex” tasks they have dealt with a single degree of freedom. In contrast, the superiority of the “recurrent architecture” was discussed specifically in the multiple degrees of freedom case (Porrill et al., Proc R Soc Lond B, 2004, Fig 3).

Secondly, I think it is a misdirection to apply correlational analysis to computational models. The authors have built the computational model; hence, they should be able to understand it mechanistically not in terms of input-output correlations. Otherwise the purpose of building a computational model is partly defeated. That is, I don’t think that it is justified to apply neuroscience-like analysis to a computational model unless that is done to reproduce experimental data or to make accurate and testable experimental predictions. Indeed, recent research has highlighted that the utility neuroscience-like analysis in enabling understanding is quite limited  ("Could a neuroscientist understand a microprocessorhttp://dx.doi.org/10.1101/055624).  All that being said, to better justify this kind of analysis and to avoid it becoming a “simulated physiology” demonstration, the authors should be more explicit in interpreting the results of this analysis. What are the predictions or practical implications? Are they generalizable? Using different types of errors will always be associated with exploiting different information on the mf pathway? I think these issues should be addressed in the second paragraph of the section 3.1.  

In the introduction, the SE and ME hypotheses are linked to Porrill et al., and Kawato's work respectively. However, according to the model in Figure 6A the authors have not confronted those two models. They have confronted the Feedback error learning with either distal-error (SE) or feedback response to error signals (ME) as teaching signal. That is, as one changes the nature of the error signal, one should change the computational architecture. To be consequent with the SE view formulated by Porrill et al. in their recurrent architecture, the output of the cerebellar should have been linked to the input of the PD module. In short, for the architecture to be consistent the nature of the teaching signal and of the output of the cerebellum should be the same. And in their case, they are only consistent in the ME case. This fact should be acknowledged in the discussion.

Finally, the authors may want to refer in the state-of-the-art to a related paper, namely, “Ruck, et al. (2016, July). Learning to Balance While Reaching: A Cerebellar-Based Control Architecture for a Self-balancing Robot. In Conference on Biomimetic and Biohybrid Systems (pp. 214-226). Springer International Publishing” as it presents a case where only sensory errors where exploited for controlling a self-balancing robot.

Author Response

First and foremost, thank you for your kind and very careful reading.
We have amended all of the specific comments and language suggestions.

MAJOR COMMENTS

Thank you for raising this important point. Testing more complex tasks (higher order of freedom) and error signals in the cerebellar model is a study we would like to pursue  in our near future study. We have not addressed it, yet, because we still need to come up with a framework for integrating the testing setup with the model. Configuring our cerebellar model for more complex tasks that include controlling objects with higher degrees of freedom may require substantial amount of changes in configuration of the model and testing set-ups, thus we think that it is beyond the scope of the current paper.   Besides, there exist infinite number of controlling objects that could be employed to test the model, and therefore it would be basically impossible to clearly conclude which error signal is the most suitable for each controlling object in our current real-world testing approach. Nonetheless, we believe that the current results, even they might be “negative evidence”,  may provide important piece of evidence to brain science, as well as the field where the artificial cerebellum is applied to control objects in real-world.  Additionally, in the view of cerebellar neuroscience, it is considered that each cerebellar zone is responsible for each axis of freedom. For example, different zones in flocculus are known to control horizontal and vertical eye movements separately (ex. Lisberger et al., J. Neurophysiol, 1994). Hence, our setup is inline with this view. To be clear about these points, we acknowledge the limitations of our results accordingly in page 15, line 370 as follows:

“The control performance with the CNN model was stable and consistent while considering a basic control plant (DC motor), and one of the most challenging plants in control engineering (the two-wheel balancing robot)[24], combined with simple and complex stimulus (low and high frequency sinusoidal desired motion) within the operation boundaries of our setup (real time operation at sampling frequency 10 ms). The characteristics of the control scenarios and our setup made difficult to reproduce the delay of 100 ms (10x the sampling frequency) shown in the visual inputs in the real cerebellum[32], which correspond to ɸ(t) and ϴ(t) in our model. However, under different circumstances, we believe our CNN model would be able to learn and produce adequate motor commands. These control plants have one and two degrees of freedom, respectively. Therefore, we may not be able to simply extend the current results to those with higher numbers of degree of freedom. There are series of evidence in cerebellar control of eye movement showing that different zones in cerebellar flocculus control different single axis of rotation of the eye (i.e. yaw and pitch rotations) [39]. Thus our current setup to control objects is inline with this view. To apply the model to controlling other objects with higher numbers of degree of freedom, we may employ the same number of cerebellar models for controlling each axis of motion. Evaluating nature of error signals suitable for this situation is in our scope of future studies.”

All that being said, to better justify this kind of analysis and to avoid it becoming a “simulated physiology” demonstration, the authors should be more explicit in interpreting the results of this analysis. What are the predictions or practical implications? Are they generalizable? Using different types of errors will always be associated with exploiting different information on the mf pathway? I think these issues should be addressed in the second paragraph of the section 3.1

Regarding the correlation study, we completely agree with the reviewer’s concern about the use of this method in a computational model. We can indeed check mechanistically the inputs/outputs of the different cells in the model, as we have demonstrated in a different manuscript with the same model (Pinzon-Morales R.D, & Hirata, Y, Front. Neural Circuit, 8, 2014), however, at this moment we would like to show a rough measure of change between different mossy fibers and granule cell activities that could possibly be observed in a modern experiment. We recognize that challenges that pose the recording of GC activity at the moment, and further technological development would open the doors to more direct studies of the input/output relationship.

To be clear about this point, the following lines have been added to the section 4.1 in the updated version of our manuscript, which corresponds to the section 3.1 in the original version.

(p. 14, lines 324-327)“Our experimental results showed that both SE and ME despite producing unique behavioral and neural changes in the CNN model, especially at the Purkinje cell output (Fig. 6B), are adequate error signals to govern successfully the plants in a diversity of control tasks. The practical implications of these results are that under certain experimental setups, investigating the consequences in motor learning when using motor or error information in the CF might yield positive but inconclusive results. Thus, the proper selection of control plants and stimuli that allow the dissociation of motor and control error contents is imperative.”

(p. 14, lines 349-357) “This means that to produce the Purkinje cell activities shown in Fig. 4 with SE and ME, granule cells with different mossy fiber inputs were engaged, and thus the differences in control performance. In particular, granule cells carrying error information were preferentially potentiated by sensory error and granule cells carrying desired motions were preferentially potentiated by motor error. In our setup the relation between granule cells activity and the CF content holds true for the two control plants used, however, extrapolation to other systems is not evident since changing the number of mossy fiber inputs and the size of the population of granule cells would result in a different mapping of inputs to parallel fibers, which in turn also affects the information that reaches the Purkinje cell. Clarifying this relation requires further study of the mossy fiber inputs to each granule cell and their correlation with the CF error content. For a study in this direction with a bi-hemispherical version of the CNN refer to [33].

In the introduction, the SE and ME hypotheses are linked to Porrill et al., and Kawato's work respectively. However, according to the model in Figure 6A the authors have not confronted those two models. They have confronted the Feedback error learning with either distal-error (SE) or feedback response to error signals (ME) as teaching signal. That is, as one changes the nature of the error signal, one should change the computational architecture. To be consequent with the SE view formulated by Porrill et al. in their recurrent architecture, the output of the cerebellar should have been linked to the input of the PD module. In short, for the architecture to be consistent the nature of the teaching signal and of the output of the cerebellum should be the same. And in their case, they are only consistent in the ME case. This fact should be acknowledged in the discussion.

Thank you for raising this important concern. Indeed, testing the models of Porrill et al. would require changing the architecture of the controller. In our work we followed a more general approach in which the architecture of the cerebellum follows the anatomical reports of the flocculus and only the content of the cf input is changed. We have added the following sentence to the manuscript to let the reader know this idiosyncrasy of our approach.   

(p. 4, line 305) “The structure of our CNN model and its connectivity with the control plant, the non-cerebellar pathway, and the vestibular nucleus remain the same disregarding the content of the error signal in our experiments. However, to be inline with the adaptive filter architecture proposed by Porrill et al. [19], our model would require the modification of the cerebellar structure so that the output of the Purkinje cell is projected to the input of the non-cerebellar pathway. We do not modify our cerebellar structure and maintain the architecture of the model as reported in the flocculus and only change the CF content.”
